# Hormetic Response of Photosystem II Function Induced by Nontoxic Calcium Hydroxide Nanoparticles

**DOI:** 10.3390/ijms25158350

**Published:** 2024-07-30

**Authors:** Panagiota Tryfon, Ilektra Sperdouli, Julietta Moustaka, Ioannis-Dimosthenis S. Adamakis, Kleoniki Giannousi, Catherine Dendrinou-Samara, Michael Moustakas

**Affiliations:** 1Laboratory of Inorganic Chemistry, Department of Chemistry, Aristotle University of Thessaloniki, 54124 Thessaloniki, Greece; tryfon.giota@gmail.com (P.T.); klegia@chem.auth.gr (K.G.); samkat@chem.auth.gr (C.D.-S.); 2Institute of Plant Breeding and Genetic Resources, Hellenic Agricultural Organization-Dimitra, 57001 Thessaloniki, Greece; 3Department of Food Science, Aarhus University, 8200 Aarhus, Denmark; julietta_moustaka@food.au.dk; 4Section of Botany, Department of Biology, National and Kapodistrian University of Athens, 15784 Athens, Greece; iadamaki@biol.uoa.gr; 5Department of Botany, School of Biology, Aristotle University of Thessaloniki, 54124 Thessaloniki, Greece; moustak@bio.auth.gr

**Keywords:** inorganic nanoparticles, microwave-assisted synthesis, biostimulants, chlorophyll fluorescence imaging analysis, effective quantum yield of PSII, non-photochemical quenching, reactive oxygen species, excess excitation energy, singlet oxygen

## Abstract

In recent years, inorganic nanoparticles, including calcium hydroxide nanoparticles [Ca Ca(OH)_2_ NPs], have attracted significant interest for their ability to impact plant photosynthesis and boost agricultural productivity. In this study, the effects of 15 and 30 mg L^−1^ oleylamine-coated calcium hydroxide nanoparticles [Ca(OH)_2_@OAm NPs] on photosystem II (PSII) photochemistry were investigated on tomato plants at their growth irradiance (GI) (580 μmol photons m^−2^ s^−1^) and at high irradiance (HI) (1000 μmol photons m^−2^ s^−1^). Ca(OH)_2_@OAm NPs synthesized via a microwave-assisted method revealed a crystallite size of 25 nm with 34% w/w of oleylamine coater, a hydrodynamic size of 145 nm, and a ζ-potential of 4 mV. Compared with the control plants (sprayed with distilled water), PSII efficiency in tomato plants sprayed with Ca(OH)_2_@OAm NPs declined as soon as 90 min after the spray, accompanied by a higher excess excitation energy at PSII. Nevertheless, after 72 h, the effective quantum yield of PSII electron transport (Φ*_PSII_*) in tomato plants sprayed with Ca(OH)_2_@OAm NPs enhanced due to both an increase in the fraction of open PSII reaction centers (q*p*) and to the enhancement in the excitation capture efficiency (F*v*’/F*m*’) of these centers. However, the decrease at the same time in non-photochemical quenching (NPQ) resulted in an increased generation of reactive oxygen species (ROS). It can be concluded that Ca(OH)_2_@OAm NPs, by effectively regulating the non-photochemical quenching (NPQ) mechanism, enhanced the electron transport rate (ETR) and decreased the excess excitation energy in tomato leaves. The delay in the enhancement of PSII photochemistry by the calcium hydroxide NPs was less at the GI than at the HI. The enhancement of PSII function by calcium hydroxide NPs is suggested to be triggered by the NPQ mechanism that intensifies ROS generation, which is considered to be beneficial. Calcium hydroxide nanoparticles, in less than 72 h, activated a ROS regulatory network of light energy partitioning signaling that enhanced PSII function. Therefore, synthesized Ca(OH)_2_@OAm NPs could potentially be used as photosynthetic biostimulants to enhance crop yields, pending further testing on other plant species.

## 1. Introduction

Nanomaterials, known for their diverse applications, present innovative solutions to the critical issue of crop protection [1,2]. Inorganic nanoparticles exhibit significant potential in plant protection due to their unique physicochemical properties, which enable effective fungal control, enhance soil nutrient availability, and allow for the controlled release of antifungal agents [3,4,5]. Some nanoparticles (NPs) have been shown to be appropriate as fertilizers, contributing to higher yields of agronomically essential crops, and also as promotion mediators of plant growth and development [6]. In specific, metal-based nanoparticles enhance crop productivity by approximately 20%, while when they are used as antimicrobial agents, they reduce disease incidence by up to 50% [7]. They also reduce nutrient leaching by 30% and enhance soil carbon sequestration by 15% [7]. The need to improve plant resilience to environmental perturbations has brought NPs to the forefront as valuable tools to resolve a threatening agricultural challenge [8,9,10].

Among the different synthesized NPs, Ca-based NPs are gaining recognition as antibacterial agents against a variety of human pathogens, including both Gram-positive and -negative bacteria [11]. In agriculture, hydrated lime [Ca(OH)_2_] is widely regarded as beneficial for improving soil pH in a range of pH values [5]. Additionally, Ca NPs have been proven effective as nematicides against *Meloidogyne incognita* and *Meloidogyne javanica* by acting as pH adjusters [12]. Foliar-sprayed Ca NPs achieved disease control in strawberry plants against *Botrytis cinerea*, also inducing plant resistance against pathogens [13]. The inhibitory activity of NPs on fungi has been shown to be due mainly to reactive oxygen species (ROS)-induced oxidative stress [14]. Despite their potential as phytoprotection agents, the use of Ca NPs in plant protection has not been extensively studied [5]. Among the tested Ca NPs, calcium hydroxide nanoparticles [Ca(OH)_2_ NPs] were the most effective nematicides [11]. The high anti-bacterial and anti-fungal efficacy of Ca(OH)_2_ NPs is ROS-mediated [15]. It has been revealed that ROS formation by nanomaterials has great potential for therapy in the areas of cancer and neuropathology and depends on the size, shape, charge, and surface area of the nanomaterials [16,17,18].

Calcium (Ca) is an essential macronutrient in plants, being fundamental for plant vigor and in the regulation of photosynthesis [19,20,21]. It serves as a secondary messenger in plants and is considered a key player in enhancing plant stress tolerance, promoting growth and development [20,22,23,24]. Plastidial-localized Ca^2+^ transporters in *Arabidopsis thaliana* have an essential role in the early signaling osmotic stress responses [25]. Calcium is frequently used in agriculture as both a fertilizer and a soil amendment [26,27]. This dual functionality of calcium-based compounds paves the way for their potential use as fungicides while simultaneously fortifying plant protection [5,28]. It has been recommended for use in nano-calcium fertilizer instead of calcium chloride for improving both fruit quality and storability [29]. Notably, the European Food Safety Authority (EFSA) has recently approved calcium hydroxide [Ca(OH)_2_] for use as a fungicide on various crops [30]. Calcium hydroxide nanoparticles stand out among nanomaterials due to their distinct physical and chemical properties, such as biocompatibility, non-toxicity, ease of synthesis, and environmental friendliness [5,31].

Calcium is a vital part of the Mn_4_CaO_5_ cluster of the oxygen-evolving complex (OEC) on photosystem II (PSII), which catalyzes the water oxidation in a catalytic cycle referred to as the S-state [32,33]. Ca removal from the OEC results in structural perturbations [34], while Ca ions are fundamental for the photoprotection and repair of PSII under environmental stress [35]. Calcium cations are also necessary for the regulation of Calvin cycle enzymes [36], increasing the enzymes’ coenzyme interactions [37]. Calcium plays an essential role in regulating membrane structure and function via stabilization of the lipid bilayers, providing structural integrity to membranes [38]. Calcium in the chloroplast membrane is correlated with stromal acidification and photosynthetic inhibition [39,40,41]. Application of exogenous Ca was shown to relieve the adverse effects of environmental stresses in plants by triggering defense responses [42,43].

Improving photosynthetic efficiency to further increase crop yield achievements is recognized as a priority research issue to meet the increasing consumption of food [44,45,46,47,48]. Photosynthesis is a sustainable process for the conversion of light energy into chemical energy, and besides producing O_2_ for maintaining earth’s oxygenic atmosphere, it supplies organic compounds necessary for sustaining the life on earth [6]. Approaches to improve photosynthesis involving reshaping mechanisms, e.g., nonphotochemical quenching (NPQ) (dissipation of excess excitation energy), are still in a primary experimental stage and/or have not achieved the required results, mainly because photosynthesis is implanted in an enormous network of closely interrelated metabolic processes, which can differ between species and even cultivars [49,50].

Chlorophyll *a* fluorescence measurements have usually been employed to evaluate the photosynthetic function, particularly PSII function [51,52,53,54,55,56,57]. The absorbed light energy by the light-harvesting complexes can be used for photochemistry or dissipated throughout diverse other pathways [58,59]. The light energy which is not utilized for photochemistry or dissipated as heat can lead to reactive oxygen species (ROS) formation, such as singlet oxygen (^1^O_2_), superoxide anions (O_2_^•−^), hydroxyl radicals (OH^−^), and hydrogen peroxide (H_2_O_2_) [60,61,62]. These partially reduced or excited forms of O_2_ are involved in both oxidative damage and signaling stress responses, playing essential functions in plant cells and also in plant development, concerning various metabolic pathways [62,63,64,65,66,67,68,69,70].

Hormesis is described as an “overcompensation” response to a disruption of homeostasis and is considered an essential evolutionary adaptive strategy [71,72,73,74,75]. Hormesis is illustrated by an inverted U-shaped response curve with a low dose or short time exposure stimulation, and a high-dose or longer duration exposure inhibition [73,76]. However, a U-shaped biphasic response curve has also been observed, with a low dose or short time inhibition and a high-dose or longer time stimulation [77,78]. Hormesis response of photosynthetic function has been reported to be triggered by the NPQ mechanism [78,79].

The growing need for sustainable agricultural practices has highlighted biostimulants as essential assets for regenerative farming [80,81]. With their diverse biological functions, biostimulants significantly contribute to boosting crop growth, enhancing nutrient utilization, increasing resilience to environmental stressors, and revitalizing soil health [81]. Biostimulants like salicylic acid [82,83,84,85,86] and melatonin [87,88,89,90] have been shown to exert several positive functions in plant biotic and abiotic stress tolerance and to induce a hormetic stimulation of the PSII function by modulating the chlorophyll content and optimizing the antenna size, which resulted in enhanced effective quantum yield of PSII photochemistry (Φ*_PSII_*) [91,92,93]. The hormetic stimulation of PSII photochemistry was initiated by the NPQ mechanism that modulated ROS production, which boosted the photosynthetic function [78,93]. However, a stimulatory response detection relies highly on a study design’s strategy, including the selection of dose range and the number and exposure duration [72,78,94,95].

Calcium-based nanoparticles (Ca NPs) have been recommended as effective anti-bacterial and anti-fungal agents but their utilization as alternative fertilizers for improving plant growth has received relatively less attention. For this purpose, microwave-assisted synthesis of hydrophobic calcium hydroxide nanoparticles coated with oleylamime [Ca(OH)_2_@OAm NPs] and their subsequent physicochemical characterization was performed. Since photosynthetic function is a significant measurement of phytotoxicity and an assessment tool prior to large-scale applications, we evaluated the impact of Ca(OH)_2_@OAm NPs on photosystem II (PSII) function by employing chlorophyll fluorescence imaging technology and using tomato (*Lycopersicon esculentum* Mill.) as a model plant.

## 2. Results

### 2.1. Synthesis and Characterization of Calcium Hydroxide Nanoparticles

X-ray diffraction (XRD) analysis of the Ca(OH)_2_@OAm NPs (Appendix A) revealed sharp and well-defined peaks at angles (2θ) of approximately 18.0°, 28.7°, 34.1°, 47.1°, and 50.8°, corresponding to the (001), (100), (101), (102), and (110) planes of Ca(OH)_2_, respectively. These diffraction peaks are consistent with standard JCPDS data (#72-0156) for crystalline Ca(OH)_2_ in the portlandite phase. The crystallite size of the NPs, estimated using the Scherrer equation and based on the (101) plane, was 25 nm. The crystallinity was determined at 83%.

The Fourier-transform infrared (FT-IR) spectrum of Ca(OH)_2_@OAm NPs (Appendix A) displays characteristic peaks that confirm their chemical composition. The broad band at around 3400 cm^−1^ indicates O-H group stretching vibrations, suggesting the presence of hydroxyl groups. Peaks at 2900 and 1650 cm^−1^ correspond to C-H stretching vibrations, confirming the presence of OAm. The sharp peak at 1400 cm^−1^ indicates C-H bending, while the peak at 432 cm^−1^ corresponds to Ca-O stretching vibrations, which are characteristic of Ca(OH)_2_.

Thermogravimetric analysis (TGA) of Ca(OH)_2_@OAm NPs was conducted from 25 to 850 °C, showing three distinct stages of weight loss, as depicted in Appendix A. The first stage, around 10% weight loss, occurred up to 120 °C, due to the evaporation of physically adsorbed water and loosely bound hydroxyl groups. The second stage, also about 9% weight loss, occurred at near 300 °C, corresponding to the thermal decomposition of the OAm coating. Overall, the total weight loss observed was 34% w/w. Additionally, the DTG plot shows a positive peak, confirming the endothermic reaction associated with the transformation of Ca(OH)_2_ to CaO, which occurs simultaneously with the release of water.

Dynamic light scattering (DLS) measurements were performed to evaluate the hydrodynamic size and ζ-potential of the NPs (Appendix A). The analysis revealed that the NPs had a hydrodynamic size of 145 ± 3.5 nm with a polydispersity index (PDI) of 0.43 (Appendix A) and a ζ-potential of +4 ± 1.2 mV (Appendix A).

### 2.2. Chlorophyll Content in Tomato Leaves Sprayed with Calcium Hydroxide Nanoparticles

The chlorophyll content in tomato leaves 72 h after spraying with both 15 mg L^−1^ and 30 mg L^−1^ Ca(OH)_2_@OAm NPs decreased significantly (*p <* 0.05), compared to the water-sprayed tomato leaves (control) (Figure 1).

### 2.3. Efficiency of the Oxygen-Evolving Complex and Maximum Efficiency of Photosystem II in Tomato Leaves Sprayed with Calcium Hydroxide Nanoparticles

The maximum efficiency of PSII photochemistry (*Fv*/*Fm*) did not change after the spray with either 15 or 30 mg L^−1^ Ca(OH)_2_@OAm NPs at both time measurements (90 min and 72 h) compared to the water-sprayed tomato leaves (control) (Figure 2a). In all chlorophyll fluorescence parameters, there were no statistically significant differences in the control leaves (water sprayed) between 90 min and 72 h measurements, and thus, the values of water-sprayed control leaves are presented as a mean value.

The efficiency of the oxygen-evolving complex (OEC) increased significantly (*p <* 0.05) 90 min after spraying with 15 mg L^−1^ Ca(OH)_2_@OAm NPs, but it did not change with 30 mg L^−1^ Ca(OH)_2_@OAm NPs, compared to water-sprayed (control) leaves (Figure 2b). Seventy-two hours after spraying with either 15 mg L^−1^ or 30 mg L^−1^ Ca(OH)_2_@OAm NPs, there was no difference in the efficiency of the OEC, compared to that in the control tomato leaves (Figure 2b).

### 2.4. Light Energy Use Efficiency and Electron Transport Efficiency in Tomato Leaves Sprayed with Calcium Hydroxide Nanoparticles

The effective quantum yield of PSII photochemistry (Φ*_PSII_*) did not change 90 min after the spray with either 15 mg L^−1^ or 30 mg L^−1^ Ca(OH)_2_@OAm NPs, compared to water-sprayed (control) leaves (Figure 3a). However, 72 h after the spray, Φ*_PSII_* increased significantly for both 15 mg L^−1^ and 30 mg L^−1^ Ca(OH)_2_@OAm NPs, compared to that of the water-sprayed (control) leaves (Figure 3a). A similar response pattern was noticed in the electron transport rate (ETR) (Figure 3b).

The non-regulated energy loss in PSII (Φ*_NO_*) at the growth irradiance (GI, 580 μmol photons m^−2^ s^−1^) did not change after the spray with either 15 mg L^−1^ or 30 mg L^−1^ Ca(OH)_2_@OAm NPs at both time measurements (90 min and 72 h), compared to that in the water-sprayed tomato leaves (control) (Figure 4a). However, 90 min after the spray, Φ*_NO_* was lower in tomato leaflets sprayed with 15 mg L^−1^ Ca(OH)_2_@OAm NPs compared to that in the 30 mg L^−1^ ones (Figure 4a).

At the high irradiance (HI, 1000 μmol photons m^−2^ s^−1^), Φ*_NO_* 90 min after the spray did not change with the 15 mg L^−1^ Ca(OH)_2_@OAm NPs but increased with the 30 mg L^−1^ Ca(OH)_2_@OAm NPs (Figure 4b). An increased Φ*_NO_* was recorded 72 h after the spray for both 15 mg L^−1^ and 30 mg L^−1^ Ca(OH)_2_@OAm NPs (Figure 4b).

### 2.5. Impact of Calcium Hydroxide Nanoparticles on Non-Photochemical Quenching and the Fraction of Open PSII Reaction Centers

Non-photochemical quenching (NPQ), which is a photoprotective mechanism, did not change after the spray with either 15 or 30 mg L^−1^ Ca(OH)_2_@OAm NPs at the GI compared to that in water-sprayed (control) leaves (Figure 5a). However, 72 h after the spray, NPQ decreased significantly at the HI, especially with 15 mg L^−1^ Ca(OH)_2_@OAm NPs (Figure 5b).

The portion of open PSII rection centers (RCs) (q*p*), which also represents the redox state of quinone A (Q_A_), did not change after the spray with either 15 or 30 mg L^−1^ Ca(OH)_2_@OAm NPs at the GI, compared to that in water-sprayed (control) leaves (Figure 6a). However, at 72 h after the spray with either 15 or 30 mg L^−1^ Ca(OH)_2_@OAm NPs, the fraction of open PSII RCs was significant higher compared to that 90 min after the spray (Figure 6a). At the HI, 90 min after the spray, the redox state of Q_A_ became more reduced (lower fraction of open PSII RCs) with the 30 mg L^−1^ Ca(OH)_2_@OAm NPs, but at 72 h after the spray, a more oxidized state of Q_A_ was observed, especially with 15 mg L^−1^ Ca(OH)_2_@OAm NPs (Figure 6b).

### 2.6. Impact of Calcium Hydroxide Nanoparticles on the Efficiency of PSII Reaction Centers

The efficiency of the open PSII RCs (F*v*’/F*m*’) did not change after the spray with either 15 or 30 mg L^−1^ Ca(OH)_2_@OAm NPs at the GI, compared to that in water-sprayed (control) leaves (Figure 7a). However, at the HI, at 72 h after the spray, an increased efficiency of the open PSII RCs (F*v*’/F*m*’) was noticed, especially with 15 mg L^−1^ Ca(OH)_2_@OAm NPs (Figure 7b).

### 2.7. Impact of Calcium Hydroxide Nanoparticles on the Spatiotemporal Heterogeneity of PSII Photochemistry

Whole tomato leaf color-coded pictures of Φ*_PSII_*, Φ*_NPQ,_* Φ*_NO_*, NPQ/4, and q*p*, obtained at the GI (580 μmol photons m^−2^ s^−1^), are presented in Figure 8. The higher leaf heterogeneity was observed in the parameters NPQ/4, q*p*, and Φ*_PSII_* 90 min after spraying with 15 mg L^−1^ (Figure 8b) or with 30 mg L^−1^ (Figure 8c) Ca(OH)_2_@OAm NPs. The slight non-significant decrease in the quantum yield of PSII photochemistry (Φ*_PSII_*) 90 min after the spray with either 15 or 30 mg L^−1^ Ca(OH)_2_@OAm NPs compared to the control (Figure 3a and Figure 8a–c), was restored after 72 h by a significant increase in Φ*_PSII_* (Figure 3a and Figure 8d,e). This increase in Φ*_PSII_* 72 h after the spray was due to the decreased NPQ compared to that in the controls (Figure 5a and Figure 8a,d,e). Thus, the lowest NPQ values observed 72 h after the spray with 15 mg L^−1^ Ca(OH)_2_@OAm NPs (Figure 8d) were accompanied by the highest Φ*_PSII_* values (Figure 8d).

In Figure 9, the leaf color-coded pictures of Φ*_PSII_*, Φ*_NPQ,_* Φ*_NO_*, NPQ/4, and q*p* obtained at the HI (1000 μmol photons m^−2^ s^−1^) are presented. The higher leaf heterogeneity was observed mainly in the parameters NPQ/4, q*p*, and Φ*_PSII_* 90 min after the spray with 30 mg L^−1^ Ca(OH)_2_@OAm NPs (Figure 9c). The increase in Φ*_PSII_* 72 h after the spray was due to the decreased NPQ compared to that in the control (Figure 9a,d,e). The lowest NPQ values observed 72 h after the spray with 15 mg L^−1^ Ca(OH)_2_@OAm NPs (Figure 9d) were accompanied by the highest Φ*_PSII_* and q*p* values (Figure 9d).

### 2.8. Impact of Calcium Hydroxide Nanoparticles on Hydrogen Peroxide Production

Hydrogen peroxide (H_2_O_2_) production in tomato leaflets was observed as a light green color in the leaf veins (Figure 10). Thirty minutes after the treatments, the highest hydrogen peroxide (H_2_O_2_) production was observed with 30 mg L^−1^ Ca(OH)_2_@OAm NPs (Figure 10c), while the lowest was with 15 mg L^−1^ (Figure 10b). Ninety minutes after the treatments, the 15 mg L^−1^ Ca(OH)_2_@OAm NPs showed lower H_2_O_2_ production (Figure 10e) compared to the 30 mg L^−1^ ones (Figure 10f), while 72 h after the spray, the higher H_2_O_2_ production was observed with 15 mg L^−1^ Ca(OH)_2_@OAm NPs (Figure 10h).

### 2.9. Hormetic Responses of Photosystem II in Tomato Leaves Sprayed with Calcium Hydroxide Nanoparticles

The response of the quantum yield of PSII photochemistry (Φ*_PSII_*) to 15 mg L^−1^ (Figure 11a) or 30 mg L^−1^ (Figure 11b) Ca(OH)_2_@OAm NPs showed an inverted J-shaped biphasic response curve. The inverted J-shaped biphasic response curve of Φ*_PSII_* to 15 mg L^−1^ Ca(OH)_2_@OAm NPs was almost identical at the GI (580 μmol photons m^−2^ s^−1^), and at the HI (1000 μmol photons m^−2^ s^−1^) (Figure 11a). The decrease in Φ*_PSII_* for more than 90 min after the spray with 15 mg L^−1^ Ca(OH)_2_@OAm NPs, (Figure 11a) was restored a little before 72 h of the spray, almost at the same time at the GI and the HI. In contrast, the Φ*_PSII_* reduction after the spray with 30 mg L^−1^ Ca(OH)_2_@OAm NPs was restored earlier at the GI than at the HI, in which it was done at 72 h after the spray (Figure 11b).

The response of the quantum yield of regulated non-photochemical energy loss in PSII (Φ*_NPQ_*) to 15 mg L^−1^ (Figure 12a) or 30 mg L^−1^ (Figure 12b) Ca(OH)_2_@OAm NPs showed a J-shaped biphasic response curve. Φ*_NPQ_* increased by both 15 and 30 mg L^−1^ Ca(OH)_2_@OAm NPs but decreased at both treatments, much earlier with 15 mg L^−1^ Ca(OH)_2_@OAm NPs (Figure 12a) compared to 30 mg L^−1^ (Figure 12b).

The response of the excess excitation energy at PSII (EXC) to Ca(OH)_2_@OAm NPs, corresponds to a J-shaped hormetic curve, with an enhancement of the EXC 90 min after the spray for both 15 (Figure 13a) and 30 mg L^−1^ (Figure 13b), Ca(OH)_2_@OAm NPs. However, long before the 72 h spray, the EXC decreased at both GI and HI for both 15 (Figure 13a) and 30 mg L^−1^ (Figure 13b), Ca(OH)_2_@OAm NPs.

## 3. Discussion

In our study, the synthesized Ca(OH)_2_@OAm NPs exhibited smaller crystallite and hydrodynamic sizes, along with a different ζ-potential compared to other studies utilizing various synthetic methods or experimental conditions [5,11,96,97]. This variability in surface charge, influenced by different surface coatings, has been mentioned in studies on Ca-based NPs [12] and tannic acid-coated Ca(OH)_2_ NPs [98]. Despite the synthetic route and organic coating, the NaOH/CaCl_2_ molar ratio also significantly influences the properties and applications of these NPs, demonstrating their efficacy in agriculture [4,97,99]. The TGA curve displays multiple weight loss steps, confirming the presence of OAm as the surface capping agent [100]. The weight loss observed in the TGA reflects the thermal decomposition of OAm, highlighting its role in stabilizing the nanoparticles through non-covalent interactions and its subsequent breakdown at elevated temperatures [101].

Through the process of photosynthesis, plants use energy from the sun to drive primary production, and thus, the light energy use efficiency is generally recognized to govern crop yields [44,45,46,47,102,103]. Designing plants with lower chlorophyll content and smaller chlorophyll antenna sizes reduces the excess absorption of sunlight and improves photosynthetic efficiency [44,91,104,105,106,107,108,109]. Rice mutants with reduced chlorophyll content presented an enhanced electron transport rate and net photosynthetic rate than their wild type [110]. Reduced chlorophyll content is correlated to smaller chlorophyll antenna sizes and reduced non-photochemical quenching (NPQ), but to increased PSII efficiency [111,112]. Seventy-two hours after the spray with Ca(OH)_2_@OAm NPs, the tomato plants with reduced chlorophyll content (Figure 1) showed an enhanced electron transport rate (ETR) (Figure 3b). The enhanced effective quantum yield of PSII electron transport (Φ*_PSII_*) (Figure 3a) was due to both an increase in the fraction of open PSII reaction centers (q*p*) (Figure 6a) and to the enhancement of the excitation capture efficiency by these centers (F*v*’/F*m*’) (Figure 7a).

The decreased NPQ 72 h after the spray with Ca(OH)_2_@OAm NPs provided evidence of the photoprotection offered by the reduced chlorophyll content and the smaller chlorophyll antenna size [111,112], especially at HI with 15 mg L^−1^ Ca(OH)_2_@OAm NPs (Figure 5b). At the same time the fraction of open PSII reaction centers (q*p*) increased (Figure 6b). The photoprotective mechanism of NPQ is considered efficient if q*p* does not decrease and if it can be at least equal to that of non-stressed plants [113,114,115,116]. Otherwise, an imbalance between absorbed light energy and its use appears, specifying excess excitation energy [113,117,118]. The redox state of quinone A (Q_A_) is recognized to be essential for retrograde signaling [116,119,120,121]. A more oxidized Q_A_, as observed 72 h after the spray with Ca(OH)_2_@OAm NPs (Figure 6a,b), corresponds to decreased stomatal opening [92,121,122] and reduced water loss for each mol of CO_2_ fixed [123]. It has been suggested that stomatal opening is not coordinated by the quantity of CO_2_ or the Calvin–Benson–Bassham cycle but instead by the redox state of Q_A_ [124]. In contrast to our results, foliar-application of 100 mg L^−1^ Ca NPs increased NPQ under drought stress conditions in hydroponically grown *Brassica napus* plants [125]. It seems that a differential photoprotective response mechanism under non-stress, mild stress, or severe stress conditions exists, as was suggested for celery plants sprayed with salicylic acid under gradual water deficit stress [86].

The increased ETR 72 h after the spray with Ca(OH)_2_@OAm NPs (Figure 3b) was due to the decreased NPQ [93,126], as we also observed (Figure 5a). However, NPQ prevents ROS development from operating as a photoprotective mechanism [58,62,118,127,128,129]. ROS, such as singlet-excited oxygen (^1^O_2_), superoxide anion radical (O_2_^•−^), and hydrogen peroxide (H_2_O_2_), are regularly created in cells but are scavenged by the antioxidant cellular mechanisms [60,61,62,93,130,131]. Although ROS can be detrimental, they also act as messengers controlling plant growth and development as well as stress responses [86,132,133]. The role of chloroplast antioxidants is not to totally delete ROS, but rather to achieve a suitable balance between generation and elimination so to complement them with photosynthesis and accomplish an efficient spread of the signal wave [93,134,135,136]. ROS are important signaling molecules that permit cells to react quickly to miscellaneous alterations of their homeostasis, establishing defense mechanisms and plant resilience [132,137,138].

Among reactive oxygen species (ROS), singlet oxygen (^1^O_2_) and hydrogen peroxide (H_2_O_2_) play key roles in initiating various signaling networks when photosynthesis is disrupted [132,135,139]. Due to its high reactivity, ^1^O_2_ initiates signaling but does not propagate it [140,141]. In contrast, H_2_O_2_, with its lower reactivity, serves as a mobile messenger within a spatially defined signaling pathway [91,135,139]. Hydrogen peroxide produced in the leaves tends to accumulate preferentially in the bundle sheath cells of leaves (Figure 10) [62,139,142,143,144,145]. Hydrogen peroxide can travel through the leaf veins, functioning as a long-distance signaling molecule [91,146].

The decreased NPQ 72 h after the spray with Ca(OH)_2_@OAm NPs (Figure 5b) was accompanied by increased Φ*_NO_* generation (Figure 4b). An increased Φ*_NO_* is regarded to be related to an increased amount of singlet oxygen (^1^O_2_) generation [147,148,149,150]. When chlorophyll molecules absorb light energy, they are transformed to the singlet-state chlorophyll (^1^Chl*) molecules. ^1^Chl* molecules can be de-excited either via the NPQ mechanism, that is, by losing energy as heat; by the process of photochemistry (q*p*); or, finally, by re-emitting light from the lowest excited state through fluorescence [58,62,127,128]. However, if ^1^Chl* molecules are not de-excited, the lower-energy triplet-state chlorophyll molecules (^3^Chl*) are formed, which can remain excited for longer periods of time and can react with molecular O_2_ to produce ^1^O_2_ [59,62,151]. Consequently, a decrease in NPQ (Figure 5b) results in increased ^1^O_2_ generation (Figure 4b). Thus, it can be suggested that NPQ can regulate, to an extent, the level of ROS in plant cells [78,152,153]. An increased H_2_O_2_ production at 90 min after the spray with 30 mg L^−1^ Ca(OH)_2_@OAm NPs (Figure 10f), compared to that in the control and 15 mg L^−1^ groups, was also accompanied by an increased Φ*_NO_* (Figure 4a,b). The lower H_2_O_2_ production at 90 min after the spray with 15 mg L^−1^ Ca(OH)_2_@OAm NPs (Figure 10e) compared to 30 mg L^−1^ (Figure 10f) matches with the higher NPQ values (Figure 5a,b, Figure 8b and Figure 9b) and the lowest Φ*_NO_* generation (Figure 4a,b, Figure 8b and Figure 9b). At 72 h after the spray, the higher H_2_O_2_ production with 15 mg L^−1^ Ca(OH)_2_@OAm NPs (Figure 10h) compared to that with 30 mg L^−1^ ones (Figure 10i) matches with the lowest NPQ values (Figure 8d and Figure 9d) and the highest Φ*_NO_* generation (Figure 8d and Figure 9d). An increased ROS generation was observed as soon as 30 min after the treatment with the 30 mg L^−1^ Ca(OH)_2_@OAm NPs (Figure 10c), while the increased ROS accumulation with 15 mg L^−1^ Ca(OH)_2_@OAm NPs was noticed 72 h after spraying (Figure 10h). The anti-bacterial and anti-fungal efficacy of Ca(OH)_2_ NPs is due to this increased ROS generation [15].

## 4. Materials and Methods

### 4.1. Synthesis of Calcium Hydroxide Nanoparticles [Ca(OH)_2_@OAm NPs]

Chemicals and Reagents: The following analytical-grade chemicals and reagents were used as received without further purification: calcium chloride (CaCl_2_, BDH Laboratory ACS, VWR Chemicals BDH^®^, Darmstadt, Germany, M = 110.9 g/mol), oleylamine (OAm, Merck, Darmstadt, Germany, M = 267.493 g/mol), sodium hydroxide (NaOH, Merck, Darmstadt, Germany, M = 39.997 g/mol), and chloroform (CHCl_3_, VWR Chemicals BDH^®^, Darmstadt, Germany, ≥99.8%).

The one-pot synthesis of Ca(OH)_2_@OAm NPs was synthesized based on a microwave-assisted approach and our previous study [5] with a modification in the molar ratio of NaOH and CaCl_2_. Initially, 10 mL of a 3 M NaOH aqueous solution was incrementally added to 0.8 g of anhydrous CaCl_2_, which had been dissolved in 30 mL of OAm with vigorous stirring. This solution was stirred consistently at 35 °C for 15 min and was transferred into a Teflon vessel. The reaction took place at 190 °C, with a 30-min hold time and a 15-min ramp-up period, using a MARS 6-240/50-CEM microwave reactor. After the microwave treatment, the autoclave was allowed to cool to room temperature naturally (Figure 14). The resulting mixture was centrifuged at 5000 rpm for 20 min and washed with CHCl_3_ to remove unwanted impurities and precursors. The reaction yield was calculated as 78%, based on the initial metal precursor and the metal content incorporated into the NPs (Figure 14).

### 4.2. Characterization of Ca(OH)_2_@OAm NPs

Physicochemical characterization of the Ca(OH)_2_@OAm NPs was carried out by various analytical techniques. X-ray diffraction (XRD) analysis was performed and the XRD patterns were acquired using a Rigaku Ultima+ X-ray diffractometer (Rigaku Corporation, Shibuya-Ku, Tokyo, Japan) with a Cu-Kα radiation source (λ = 1.541 Å) operating at 40 kV/30 mA. The crystallite size was estimated using the Scherrer equation, while the crystallinity of the NPs was calculated based on the methodology described by Khan et al. (2019). Fourier-transform infrared (FT-IR) spectroscopy was performed using a Nicolet iS20 FT-IR spectrometer (Thermo Fisher Scientific, Waltham, MA, USA) equipped with a monolithic diamond, attenuated total reflection (ATR) crystal. Thermal properties were analyzed using a SETARAM SetSys-1200 (KEP Technologies, Caluire, France) instrument. Differential Thermogravimetric (DTG) and Thermogravimetric Analysis (TGA) measurements were performed under a nitrogen atmosphere, with a heating rate of 10 °C/min. Dynamic light scattering (DLS) analysis was used to assess the hydrodynamic size (nm), polydispersity index (PDI), and ζ-potential (mV) of the NPs solution at 25 °C, employing a Zetasizer Nano ZS Malvern apparatus (VASCO Flex™ Particle Size Analyzer NanoQ V2.5.4.0, Northampton, UK).

### 4.3. Plant Material and Growth Conditions

Tomato (*Lycopersicon esculentum* Mill. cv Galli) plants were purchased from the market in pots and left for three days in a greenhouse at 25 ± 1/20 ± 1 °C, day/night temperature, with 65 ± 5/75 ± 5% day/night relative humidity and 14 h day/night photoperiod provided by a photosynthetic photon flux density (PPFD) of 580 ± 10 μmol quanta m^−2^ s^−1^.

### 4.4. Exposure of Plants to Ca(OH)_2_@OAm NPs

The tomato plants after the acclimation period were foliar-sprayed once either with 15 mL of distilled water (control), 15 mg L^−1^ Ca(OH)_2_@OAm NPs, or 30 mg L^−1^ Ca(OH)_2_@OAm NPs. Three plants were used in each treatment with three independent replicates.

### 4.5. Measurements of Chlorophyll Content

Chlorophyll content of all treated tomato plants was measured photometrically with the chlorophyll content meter (Model Cl-01, Hansatech Instruments Ltd., Norfolk, UK) [154] and is given in relative units.

### 4.6. Chlorophyll Fluorescence Imaging Analysis

Chlorophyll *a* fluorescence parameters that correlated to PSII function were measured using the modulated Imaging-PAM Fluorometer M-Series (Heinz Walz GmbH, Effeltrich, Germany) as explained in detail previously [155]. Tomato plants, after being sprayed with either 15 mg L^−1^ Ca(OH)_2_@OAm NPs, 30 mg L^−1^ Ca(OH)_2_@OAm NPs, or distilled water (control), were dark-adapted for 30 min, and then chlorophyll fluorescence measurements were contacted 90 min and 72 h after the spray. The actinic light (AL) intensities used were 580 μmol photons m^−2^ s^−1^ (growth light, GL) and 1000 μmol photons m^−2^ s^−1^ (high light, HL). The definitions of the chlorophyll fluorescence parameters calculated using the V2.41a Win software (Heinz Walz GmbH, Effeltrich, Germany) are described in Appendix A. Color-coded whole leaf images of some parameters are also presented.

### 4.7. Evaluation of Hydrogen Peroxide Production

Hydrogen peroxide (H_2_O_2_) generation in tomato leaflets was evaluated 30 min, 90 min, and 72 h after tomato plants were sprayed with either 15 mg L^−1^ Ca(OH)_2_@OAm NPs, 30 mg L^−1^ Ca(OH)_2_@OAm NPs, or distilled water (control), as described previously [140]. Leaves of all treated tomato plants were incubated with 25 μM 2′,7′-dichlorofluorescein diacetate (DCF-DA, Sigma Aldrich, Chemie GmbH, Schnelldorf, Germany) for 30 min in the dark and then observed with a Zeiss AxioImager Z2 epi-fluorescence microscope and photographed with an AxioCam MRc5 digital camera [143].

### 4.8. Statistical Analysis

All statistical analyses were performed with the version 4.3.1 R software (R Core Team, 2023). Normality and homogeneity of variance was tested with the Shapiro–Wilk test and Levene’s test. Subsequently, a two-way ANOVA was accomplished for each photosynthetic parameter with Treatment (control, 15 mg L^−1^ Ca(OH)_2_@OAm NPs, or 30 mg L^−1^ Ca(OH)_2_@OAm NPs) and Time (90 min and 72 h) as factors, followed by post hoc analysis with Tukey’s honest significant difference method with the R package ‘multcomp’. Values at *p* < 0.05 were considered significantly different.

## 5. Conclusions

Oleylamine-coated Ca(OH)_2_ nanoparticles [Ca(OH)_2_@OAm NPs] were efficiently synthesized, resulting in small size and high crystallinity. Seventy-two hours after the spray of tomato plants with Ca(OH)_2_@OAm NPs, the light absorbed by tomato leaves with the reduced chlorophyll content was more efficiently partitioned to photochemistry, signifying an improved photosynthetic efficiency. The hormetic response of the quantum yield of PSII photochemistry was much earlier with the 15 mg L^−1^ Ca(OH)_2_ NPs than with the 30 mg L^−1^. It is suggested that the hormetic stimulation of PSII functionality was triggered by the non-photochemical quenching (NPQ) mechanism that stimulated ROS production, which enhanced the photosynthetic function. It can be concluded that calcium hydroxide nanoparticles [Ca(OH)_2_@OAm NPs] by effectively regulating NPQ mechanism, enhanced the electron transport rate and decreased the excess excitation energy in tomato seedlings. However, there are limited studies specifically addressing the synthesis of oleylamine-coated Ca NPs, highlighting the need for further research in this area. Foliar applied nano-fertilizers not only can control nutrient release, enhancing nutrient use efficiency (NUE) [156], but can also stimulate photosynthetic function [157] and alleviate the adverse effects of abiotic stresses [158].

## Figures and Tables

**Figure 1 ijms-25-08350-f001:**
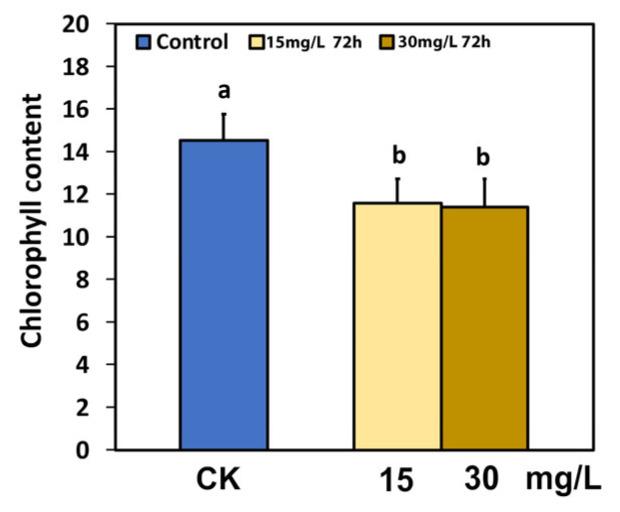
The chlorophyll content of tomato leaflets 72 h after spraying with water (control) or 15 and 30 mg L^−1^ Ca(OH)_2_@OAm NPs. Significant statistical difference (*p* < 0.05) is shown by different lowercase letters. Error bars in columns are SDs.

**Figure 2 ijms-25-08350-f002:**
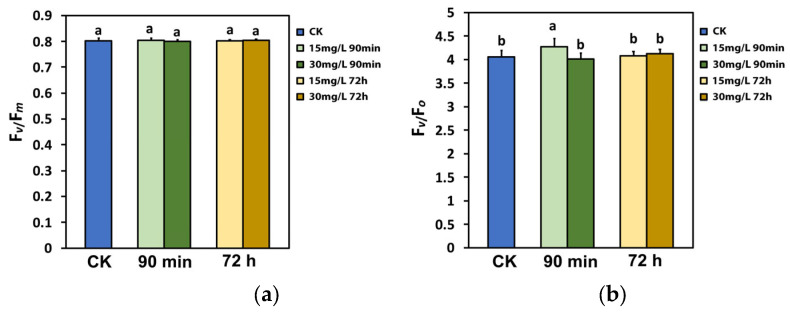
The maximum efficiency of PSII photochemistry (*Fv*/*Fm*) and (**a**) the efficiency of the oxygen-evolving complex (OEC) (*Fv*/*Fo*) (**b**) of tomato leaflets, 90 min and 72 h after spraying with water (control), or 15 mg L^−1^ and 30 mg L^−1^ Ca(OH)_2_@OAm NPs. Significant statistical difference (*p* < 0.05) is represented by different lowercase letters. SD is shown as error bar in the columns.

**Figure 3 ijms-25-08350-f003:**
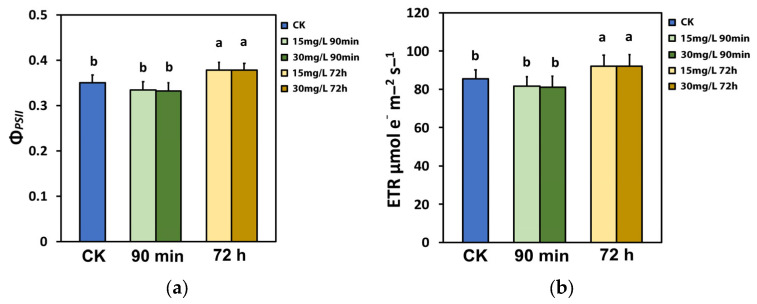
The effective quantum yield of PSII photochemistry (Φ*_PSII_*) and (**a**) the electron transport rate (ETR) (**b**) of tomato leaflets, 90 min and 72 h after spraying with water (control), or 15 mg L^−1^ and 30 mg L^−1^ Ca(OH)_2_@OAm NPs, measured at the growth irradiance (GI, 580 μmol photons m^−2^ s^−1^). Significant statistical difference (*p* < 0.05) is represented by different lowercase letters. SD is shown as error bar in the columns.

**Figure 4 ijms-25-08350-f004:**
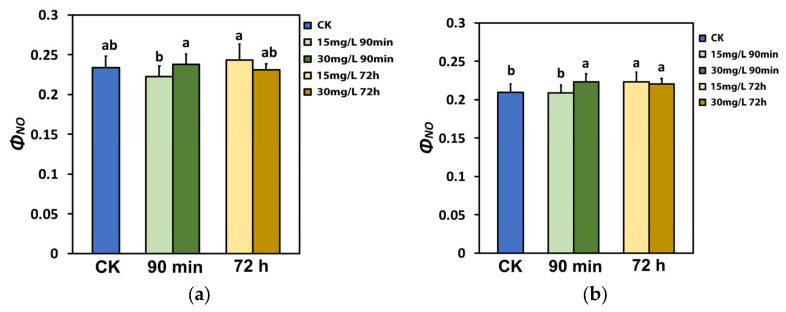
The quantum yield of non-regulated energy loss in PSII (Φ*_NO_*) at the growth irradiance (GI, 580 μmol photons m^−2^ s^−1^) (**a**) and at the high irradiance (HI, 1000 μmol photons m^−2^ s^−1^) (**b**) of tomato leaflets 90 min and 72 h after spraying with water (control) or 15 mg L^−1^ and 30 mg L^−1^ Ca(OH)_2_@OAm NPs. Significant statistical difference (*p* < 0.05) is represented by different lowercase letters. SD is shown as error bar in the columns.

**Figure 5 ijms-25-08350-f005:**
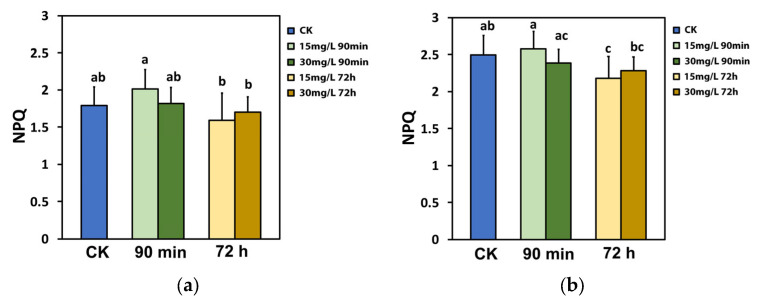
The non-photochemical quenching (NPQ) at the growth irradiance (GI, 580 μmol photons m^−2^ s^−1^) (**a**) and at the high irradiance (HI, 1000 μmol photons m^−2^ s^−1^) (**b**) of tomato leaflets, 90 min and 72 h after spraying with water (control) or 15 mg L^−1^ and 30 mg L^−1^ Ca(OH)_2_@OAm NPs. Significant statistical difference (*p* < 0.05) is presented by different lowercase letters. SD is shown as error bar in the columns.

**Figure 6 ijms-25-08350-f006:**
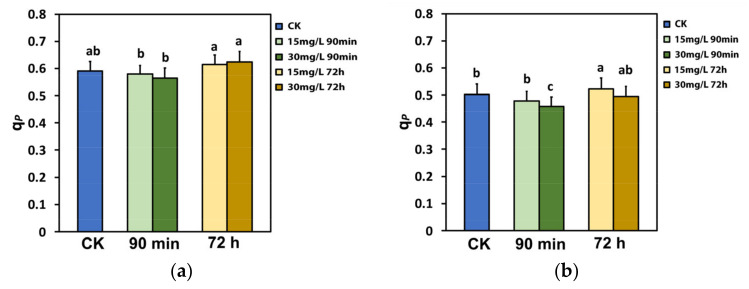
The fraction of the open PSII rection centers (RCs) (q*p*) at the growth irradiance (GI, 580 μmol photons m^−2^ s^−1^) (**a**) and at the high irradiance (HI, 1000 μmol photons m^−2^ s^−1^) (**b**) of tomato leaflets 90 min and 72 h after spraying with water (control) or 15 mg L^−1^ and 30 mg L^−1^ Ca(OH)_2_@OAm NPs. Significant statistical difference (*p* < 0.05) is represented by different lowercase letters. SD is shown as error bar in the columns.

**Figure 7 ijms-25-08350-f007:**
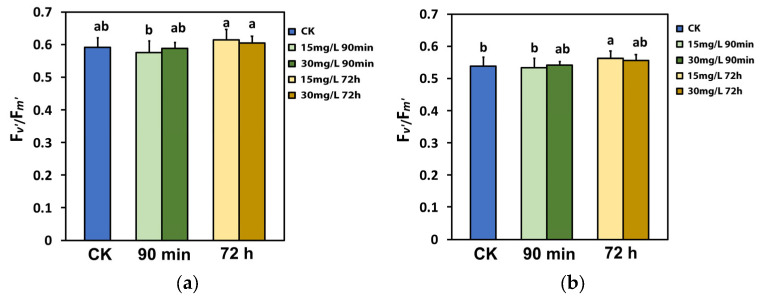
The efficiency of the open PSII RCs (F*v*’/F*m*’) at the growth irradiance (GI, 580 μmol photons m^−2^ s^−1^) (**a**) and at the high irradiance (HI, 1000 μmol photons m^−2^ s^−1^) (**b**) of tomato leaflets 90 min and 72 h after spraying with water (control) or 15 mg L^−1^ and 30 mg L^−1^ Ca(OH)_2_@OAm NPs. Significant statistical difference (*p* < 0.05) is represented by different lowercase letters. SD is shown as error bar in the columns.

**Figure 8 ijms-25-08350-f008:**
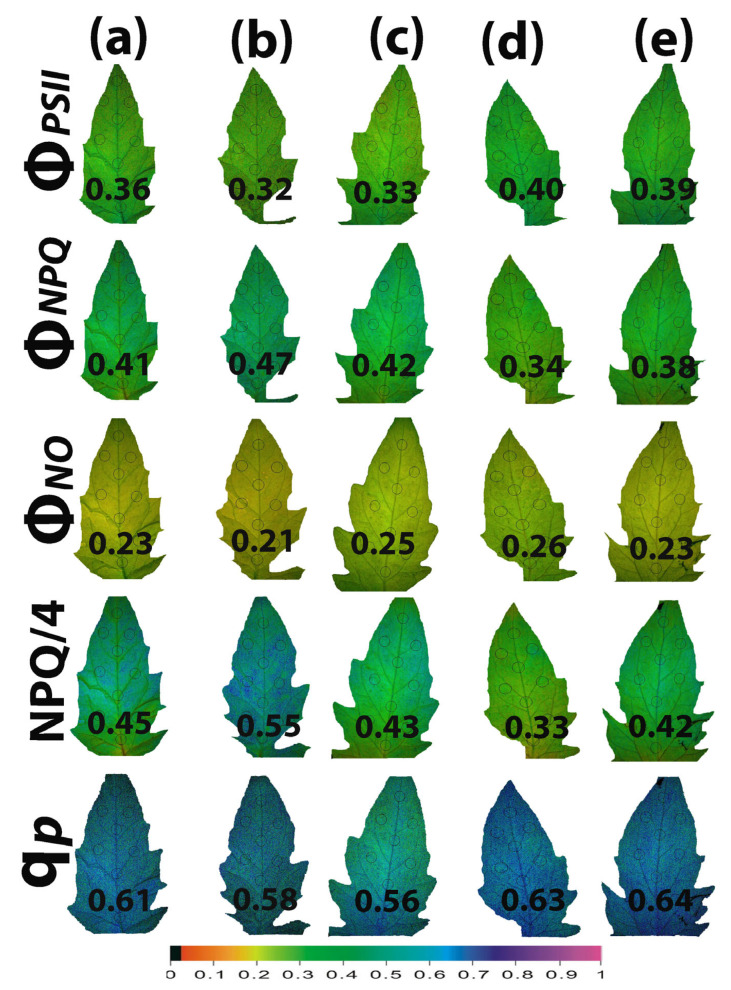
Color-coded whole leaf pictures, of Φ*_PSII_*, Φ*_NPQ,_* Φ*_NO_*, NPQ/4, and q*p*, obtained with chlorophyll fluorescence imaging at the growth irradiance (GI, 580 μmol photons m^−2^ s^−1^) at 90 min (**a**–**c**) and 72 h (**d**,**e**) after spraying with water (control) (**a**) or 15 mg L^−1^ (**b**,**d**) and 30 mg L^−1^ (**c**,**e**) Ca(OH)_2_@OAm NPs. At the bottom, the color code indicates the parameter value as color with a scale from 0 to 1.

**Figure 9 ijms-25-08350-f009:**
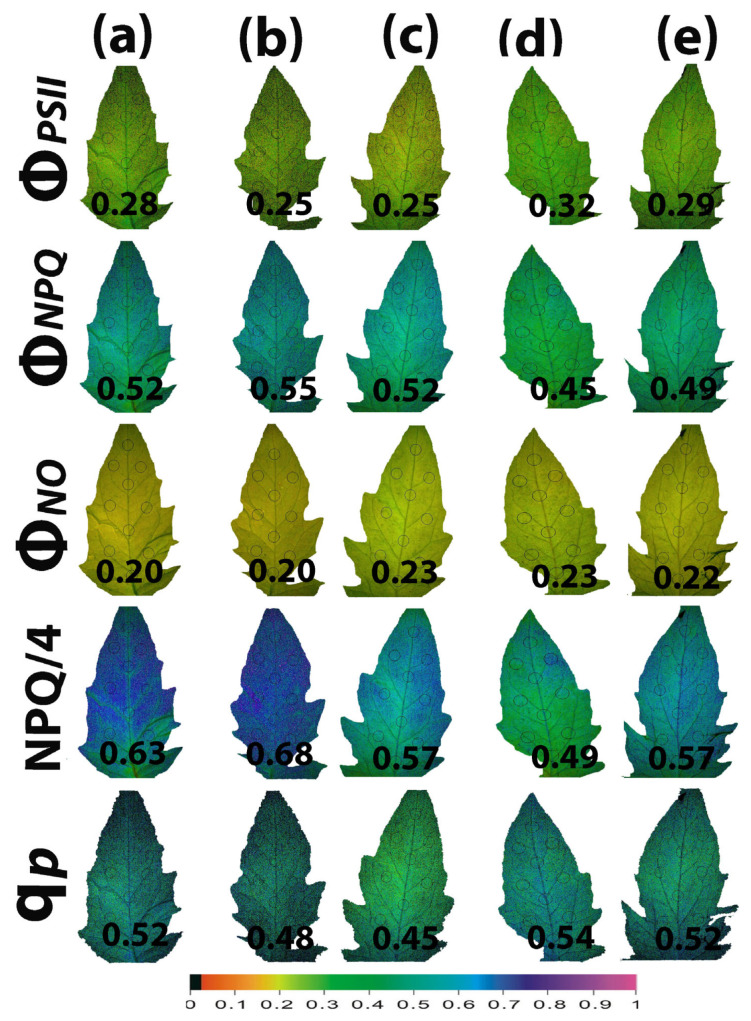
Color-coded whole leaf pictures of Φ*_PSII_*, Φ*_NPQ,_* Φ*_NO_*, NPQ/4, and q*p*, obtained with chlorophyll fluorescence imaging, at the high irradiance (HI, 1000 μmol photons m^−2^ s^−1^) at 90 min (**a**–**c**) and 72 h (**d**,**e**) after spraying with water (control) (**a**) or 15 mg L^−1^ (**b**,**d**) and 30 mg L^−1^ (**c**,**e**) Ca(OH)_2_@OAm NPs. At the bottom, the color code indicates the parameter value as color with a scale from 0 to 1.

**Figure 10 ijms-25-08350-f010:**
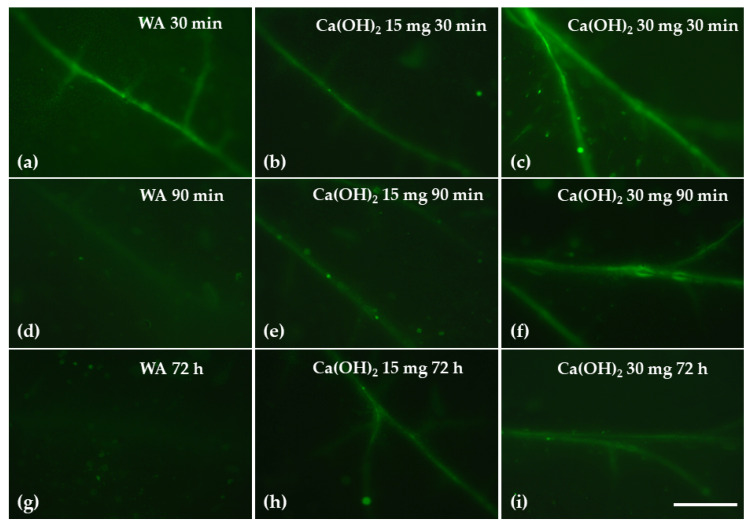
Hydrogen peroxide (H_2_O_2_) production in tomato leaflets 30 min (**a**–**c**), 90 min (**d**–**f**), and 72 h (**g**–**i**) after spraying with water (control) (**a**,**d**,**g**) or 15 mg L^−1^ (**b**,**e**,**h**) and 30 mg L^−1^ (**c**,**f**,**i**) Ca(OH)_2_@OAm NPs. The light green color denotes the H_2_O_2_ generation. Scale bar: 200 µm.

**Figure 11 ijms-25-08350-f011:**
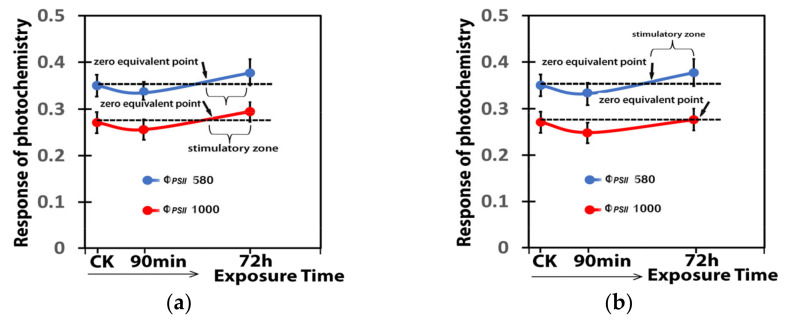
The inverted J-shaped biphasic response curve of the quantum yield of PSII photochemistry (Φ*_PSII_*) to 15 mg L^−1^ (**a**) or 30 mg L^−1^ (**b**), Ca(OH)_2_@OAm NPs, measured at the GI (580 μmol photons m^−2^ s^−1^) and at the HI (1000 μmol photons m^−2^ s^−1^), 90 min and 72 h after the spray, compared to control. The decrease in Φ*_PSII_* at the GI was restored almost at the same time for the sprays with 15 and 30 mg L^−1^ Ca(OH)_2_@OAm NPs, but at the HI, the restore in Φ*_PSII_* was much later at the 30 mg L^−1^ Ca(OH)_2_@OAm NPs.

**Figure 12 ijms-25-08350-f012:**
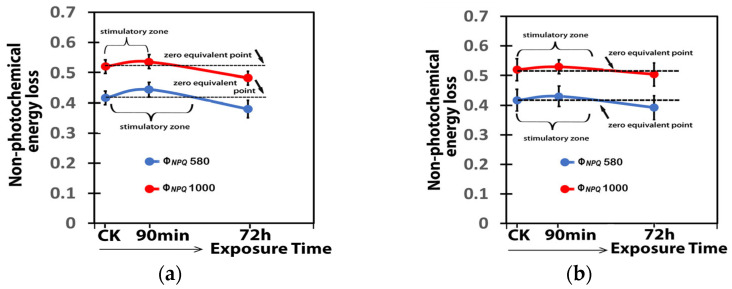
The J-shaped biphasic response curve of the quantum yield of regulated non-photochemical energy loss in PSII (Φ*_NPQ_*) to 15 mg L^−1^ (**a**) or 30 mg L^−1^ (**b**) Ca(OH)_2_@OAm NPs, measured at the GI (580 μmol photons m^−2^ s^−1^) and at the HI (1000 μmol photons m^−2^ s^−1^) 90 min and 72 h after the spray, compared to control.

**Figure 13 ijms-25-08350-f013:**
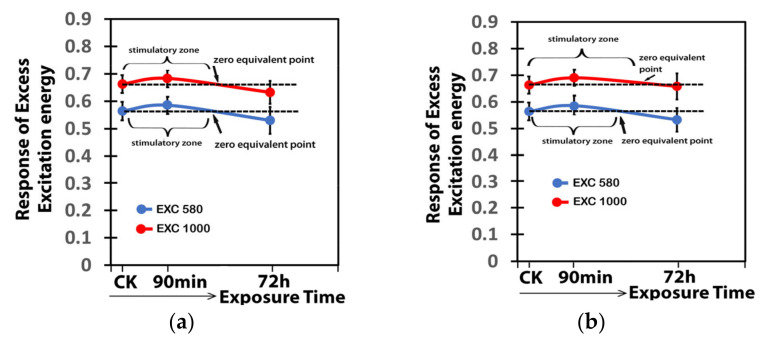
The J-shaped biphasic response curve of the excess excitation energy PSII (EXC) to 15 mg L^−1^ (**a**) or 30 mg L^−1^ (**b**) Ca(OH)_2_@OAm NPs, measured at the GI (580 μmol photons m^−2^ s^−1^) and at the HI (1000 μmol photons m^−2^ s^−1^) 90 min and 72 h after the spray, compared to control. The decrease in the EXC after the spray with the 15 mg L^−1^ Ca(OH)_2_@OAm NPs occurred almost at the same time for the GI and the HI, but with the 30 mg L^−1^ Ca(OH)_2_@OAm NPs, the decrease in the EXC occurred earlier at the GI compared to the HI.

**Figure 14 ijms-25-08350-f014:**
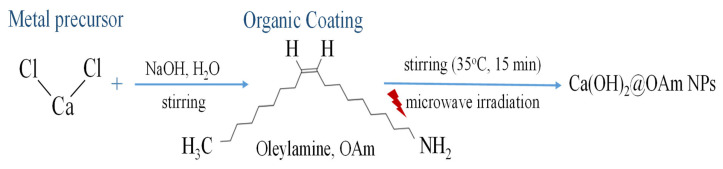
A schematic representation of the synthesis of Ca(OH)_2_@OAm NPs.

## Data Availability

The data presented in this study are available in this article.

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
