# Peer review of "Hormetic Response of Photosystem II Function Induced by Nontoxic Calcium Hydroxide Nanoparticles"

_ijms, 2024, doi:10.3390/ijms25158350_

Round 1

Reviewer 1 Report

Comments and Suggestions for Authors

Dear Authors, I have reviewed the manuscript and provide my comments below:

The theme of the manuscript is that synthesized Ca(OH)2@OAm NPs can potentially be used as photosynthetic biostimulants to increase crop yields - a prominent theme in today's agriculture in the development of sustainable agricultural practices, but some things need to be modified:

The title and abstract are appropriate, but the number of keywords is too many and it would be advisable to delete the brackets, it is not appropriate to do so. 

Introduction: basically appropriate, but there is a lack of coherence between the different topics, so it would be useful to bring them together a little better in a few sentences. Unfortunately the hypothesis at the end of the chapter is not well formulated, I suggest rewording it. 

Diagrams: why is the x axis not marked? Why is the diagram not somewhat edited?

Conclusions: it is useful to include the implications of the results for research, for agriculture in general and to highlight why this work was important globally. 

Author Response

Dear Authors, I have reviewed the manuscript and provide my comments below:

The theme of the manuscript is that synthesized Ca(OH)2@OAm NPs can potentially be used as photosynthetic biostimulants to increase crop yields - a prominent theme in today's agriculture in the development of sustainable agricultural practices, but some things need to be modified:

We would like to thank you for dedicating the time and the effort to review our work and for the valuable comments and suggestions that we adopted in our revised manuscript.

The title and abstract are appropriate, but the number of keywords is too many and it would be advisable to delete the brackets, it is not appropriate to do so.

We deleted the bracketsand omitted one keyword. The number of keywords should be according to the instructions between 3-10. We have now 9 keywords.

Introduction: basically appropriate, but there is a lack of coherence between the different topics, so it would be useful to bring them together a little better in a few sentences. Unfortunately the hypothesis at the end of the chapter is not well formulated, I suggest rewording it. 

We rearranged the paragraphs in Introduction to acquire a better coherence between the different topics. We rewrote the hypothesis.

Diagrams: why is the x axis not marked? Why is the diagram not somewhat edited?

The units of x axis were added in all Figures.

Conclusions: it is useful to include the implications of the results for research, for agriculture in general and to highlight why this work was important globally. 

 Conclusion section was expanded.

Reviewer 2 Report

Comments and Suggestions for Authors

The manuscript titled “Hormetic Response of Photosystem II Function Induced by Nontoxic Calcium Hydroxide Nanoparticles” by Tryfon, P.; et al. is a scientific work where the authors demonstrated the positive impact of calcium hydroxide nanoparticles to enhance the photosynthetic activity of tomato plants. For it many complementary techniques were devoted in this research. This is a topic of growing interest. However, it exists some points that need to be addressed (please, see them below detailed point-by-point) to improve the scientific quality of the submitted manuscript paper before this article will be consider for its publication in the International Journal of Molecular Sciences.

1) “Some nanoparticles (NPs) have been shown (…) plant growth and development” (lines 49-51). Could the authors provide quantitative data insights about the worldwide economic incidence of the agronomy sector? This will significantly aid the potential readers to better understand the significance of this devoted research.

2) “Plastidial-localized Ca2+ transporters in Arabidopsis thaliana have essential role in the early signalling osmotic stress responses (…) and environmental friendliness” (lines 57-67). Here, even if I agree with these statements furnished by the authors it should be also discussed the positive role of calcium ions in the photosystem II [1] increasing the enzyme:coenzyme interactions [2]. This will strengthen the information stated by the authors in the lines 83-91.

[1] Koua, F.H.M. Structural Changes in the Acceptor Site of Photosystem II upon Ca2+/Sr2+ Exchange in the Mn4CaO5 Cluster Site and the Possible Long-Range Interactions. Biomolecules 2019, 9, 371. https://doi.org/10.3390/biom9080371.

[2] Pérez-Domínguez, S.; et al. Nanomechanical Study of Enzyme: Coenzyme Complexes: Bipartite Sites in Plastidic Ferredoxin-NADP+ Reductase for the interaction with NADP. Antioxidants 2022, 11, 537. https://doi.org/10.3390/antiox11030537.

3) “2.1. Synthesis and characterization of Calcium Hydroxide Nanoparticles” (lines 143-170). What are the average dimenions of calcium hydroxide nanoparticles? Did the authors carry out scanning electron microscopy or dynamic light scattering measurements to unravel this information?

4) Did the authors conduct the measurements at different pH values according to those found in the soil? Some information should be furnished in this regard in order to better understand the impact shown by this parameter.

5) Discussion (lines 343-430). This section perfectly remarks the most relevant outcomes found in this work. No actions are requested from the authors.

6) “4.1. Synthesis of Calcium Hydroxide Nanoparticles [Ca(OH)2@OAm NPs]” (lines 432-449). A schematic representation of the chemical procedure to synthesize these nanoparticles should be added to illustrate the potential readers about the strategy pursued by the authors.

7) Conclusions (lines 508-520). This section unequivocally highlights the promising future perspectives opened in this field. Some quantitative details found in this research should be detailed in this section.

Author Response

The manuscript titled “Hormetic Response of Photosystem II Function Induced by Nontoxic Calcium Hydroxide Nanoparticles” by Tryfon, P.; et al. is a scientific work where the authors demonstrated the positive impact of calcium hydroxide nanoparticles to enhance the photosynthetic activity of tomato plants. For it many complementary techniques were devoted in this research. This is a topic of growing interest. However, it exists some points that need to be addressed (please, see them below detailed point-by-point) to improve the scientific quality of the submitted manuscript paper before this article will be consider for its publication in the International Journal of Molecular Sciences.

We would like to thank you for dedicating the time and the effort to review our workand for the valuable comments and suggestions that we adopted in our revised manuscript.

1) “Some nanoparticles (NPs) have been shown (…) plant growth and development” (lines 49-51). Could the authors provide quantitative data insights about the worldwide economic incidence of the agronomy sector? This will significantly aid the potential readers to better understand the significance of this devoted research.

A new sentence with such data was inserted on lines 56-59.

2) “Plastidial-localized Ca2+transporters in Arabidopsis thaliana have essential role in the early signalling osmotic stress responses (…) and environmental friendliness” (lines 57-67). Here, even if I agree with these statements furnished by the authors it should be also discussed the positive role of calcium ions in the photosystem II [1] increasing the enzyme:coenzyme interactions [2]. This will strengthen the information stated by the authors in the lines 83-91.

[1] Koua, F.H.M. Structural Changes in the Acceptor Site of Photosystem II upon Ca2+/Sr2+Exchange in the Mn4CaO5Cluster Site and the Possible Long-Range Interactions. Biomolecules2019, 9, 371. https://doi.org/10.3390/biom9080371.

[2] Pérez-Domínguez, S.; et al. Nanomechanical Study of Enzyme: Coenzyme Complexes: Bipartite Sites in Plastidic Ferredoxin-NADP+Reductase for the interaction with NADP. Antioxidants2022, 11, 537. https://doi.org/10.3390/antiox11030537.

The suggested citations were inserted and discussed.

3) “2.1. Synthesis and characterization of Calcium Hydroxide Nanoparticles” (lines 143-170). What are the average dimenions of calcium hydroxide nanoparticles? Did the authors carry out scanning electron microscopy or dynamic light scattering measurements to unravel this information?

Dynamic light scattering (DLS) measurements and ζ-potential have already been given at Lines 288-291 and were figured (Figures S4a,b).

4) Did the authors conduct the measurements at different pH values according to those found in the soil? Some information should be furnished in this regard in order to better understand the impact shown by this parameter.

Experiments were conducted only by foliar application, as it is already given in the text. In this regard measurements are independent of the soil pH values.

5) Discussion (lines 343-430). This section perfectly remarks the most relevant outcomes found in this work. No actions are requested from the authors.

Thank you for your positive comment.

 6) “4.1. Synthesis of Calcium Hydroxide Nanoparticles [Ca(OH)2@OAm NPs]” (lines 432-449). A schematic representation of the chemical procedure to synthesize these nanoparticles should be added to illustrate the potential readers about the strategy pursued by the authors.

A schematic representation of the chemical procedure to synthesize Ca(OH)2@OAm NPs was added as Figure 14 (lines 743-745). Also, a graphical abstract was given.

7) Conclusions (lines 508-520). This section unequivocally highlights the promising future perspectives opened in this field. Some quantitative details found in this research should be detailed in this section.

Some quantitative details were added in this section.

Round 2

Reviewer 1 Report

Comments and Suggestions for Authors

That is okay, thank you, I recommend it for publication